# Unsupervised Whole Object Discovery by Contextual Grouping with Repulsion

## Abstract

It is challenging to discover and segment *whole* objects from *unlabeled* images, as features unsupervisedly learned on images tend to focus on distinctive appearances (e.g., the *face* rather than the *torso*), and grouping by *feature similarity* could reveal only these representative parts, not the whole objects (e.g., the *entire human body*). Our key insight is that, *an object of distinctive parts* pops out as a whole, due not only to *how similar they are to each other*, but also to *how different they are from their contexts* within an image or across related images. The latter could be crucial for binding different parts into a coherent whole without preconception of objects. We formulate our idea for unsupervised object segmentation in a spectral graph partitioning framework, where nodes are patches and edges are grouping cues between patches, measured by feature similarity for attraction, and by feature dissimilarity for repulsion. We seek the graph cuts that maximize within-group attraction and figure-ground repulsion while minimizing figure/ground attraction and within-group repulsion. Our simple method consistently outperforms the state-of-the-art on unsupervised object discovery, figure/ground saliency detection, and unsupervised video object segmentation benchmarks. In particular, it excels at discovering whole objects instead of salient parts.

## 1 Introduction

We consider the unsupervised learning task of discovering and segmenting *whole* objects from a set of unlabeled images. Any computational model that achieves this goal is not only useful in practical applications, where segmentation annotations are tedious and costly to obtain, but also illuminating in understanding how infants make sense of their visual world from initial undivided sensations.

Existing works (Yang et al., 2019b; 2021b; Liu et al., 2021) accomplish this task by learning from unlabeled videos. AMD (Liu et al., 2021) assumes that a video contains different views of the same

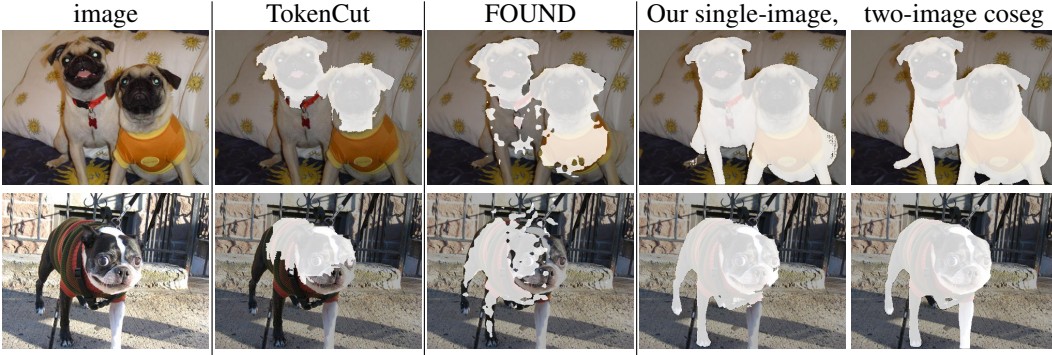

Figure 1: We propose to segment whole objects without any supervision by incorporating feature dissimilarity (repulsion) as cues. Existing methods TokenCut (Wang et al., 2023) and FOUND (Siméoni et al., 2023), which rely solely on feature similarity, often segment partial objects, like the *dog's face*, while missing other components like *legs* or *bodies*. In contrast, we capture the nexus of feature similarities and dissimilarities within and across images in a joint weighted graph. This enables the segmentation of entire objects from their backgrounds.

scene related by moving components, and the right region segmentation and region flow can be learned concurrently by image reconstruction over time. However, the performance of AMD is constraint by its piece-wise constant motion assumption within the same segment. Recent works (Wang et al., 2023; Melas-Kyriazi et al., 2022) show that objectness can be discovered from unlabeled images in attention maps of self-supervised ViT such as DINO (Caron et al., 2021). Nevertheless, features learned in such a self-supervised manner (Wu et al., 2018; Chen et al., 2020a; He et al., 2020; Misra & Maaten, 2020) tend to focus on distinctive appearances. If we cluster patches by feature similarity via e.g., TokenCut (Wang et al., 2023), we can only discover parts of characteristic appearances such as *faces*, but miss parts of plain appearances such as *torso* and *legs* of a whole object (Fig. 1).

We aim to discover *whole* instead of partial objects without any preconception of objects. Our key insight is that, *an object of distinctive parts* pops out as a whole, due not only to *how similar they are to each other*, but also to *how different they are from their contexts* within an image or across related images. The latter could be crucial for binding different parts into a coherent whole, in a bottom-up data-driven manner. For example, while the *faces* of two different dogs look similar, their *torsos* and *legs* are only mildly similar to the *faces*. However, all these parts are more dissimilar to their surrounding backgrounds. It's this common *repulsion* against the contexts they are embedded in, in addition to *attraction* of varying strengths within the objects, that helps bind object parts of heterogeneous appearances into coherent wholes.

We formulate our approach to unsupervised whole object segmentation within a spectral graph partitioning framework. In this setup, image patches are represented as nodes, and edges are the grouping relationships between these patches. These relationships are quantified using two measures: attraction based on feature similarity, and repulsion based on feature dissimilarity. The goal is to find the graph cuts that simultaneously maximize within-group attraction and between-group repulsion, while minimizing figure-to-background attraction and within-group repulsion.

We investigate this idea not only within a single image but also across related images in a co-segmentation setting, where contextual grouping with repulsion between similar images brings additional power for discovering whole objects together (Fig. 1). These images should be semantically similar yet visually distinct; *if they are identical, co-segmentation lacks new information, and if they are semantically unrelated, co-segmentation becomes ineffective*.

We present a method for unsupervised object segmentation by contextual grouping with repulsion, named CGR . With whole objectness revealed by attraction and repulsion, we further refine the self-supervised features with an attached segmentation head over the whole object masks. Our method consistently outperforms the state-of-the-art on unsupervised object discovery, unsuperivsed saliency detection, and unsupervised video object segmentation benchmarks.

Contributions. **1)** We propose to leverage contextual relationship from both within-image and cross-image to group distinctive parts into coherent whole objects without any annotations. **2)** We optimize the grouping with a new framework using attraction and repulsion cues of self-supervised ViT features. **3)** We further enhance the self-supervised ViT features by re-training the backbone with an attached segmentation head over whole object masks, thereby achieving state-of-the-art performance on multiple unsupervised segmentation benchmarks.

## 2 RELATED WORK

**Unsupervised Object Discovery.** Unsupervised object discovery aims at localizing and segmenting objects from a collection of unlabeled images. Most of current works utilize self-supervised features from visual transformers (Caron et al., 2021; Chen et al., 2020b; Caron et al., 2020). SelfMask (Shin et al., 2022) applies spectral clustering on multiple self-supervised features to extract object masks. LOST (Siméoni et al., 2021) utilizes feature similarities to localize an object seed and expands the seed to all similar patches. FreeSOLO (Wang et al., 2022) presents a FreeMask predictions from feature similarities which are ranked and filtered by a maskness score. Deep Spectral Methods (Melas-Kyriazi et al., 2022) and TokenCut (Wang et al., 2023) build a weighted graph using feature similarities (attraction) and conduct graph cut to separate objects from backgrounds. FOUND (Siméoni et al., 2023) first searches a background seed to localize objects and HEAP (Zhang et al., 2024) applies contrastive learning to learn clustered feature embeddings. PEEKABOO (Zunair & Hamza, 2024) presents to hide part of images and localize the objects with remaining image information. However, all these methods are limited in discovering whole objects as self-supervised features only capture

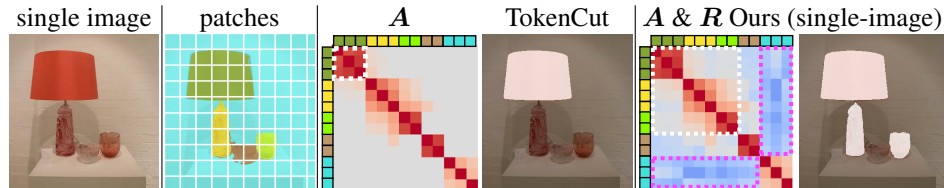

| single image | patches | $\boldsymbol{A}$ | TokenCut | $\boldsymbol{A}$ & $\boldsymbol{R}$ Ours (single-image) |

Figure 2: **We analyze feature similarities among various patches of the foreground ($f$) and the background ($g$) within a single image**. The attraction matrix ($\boldsymbol{A}$) reveals that certain parts of foreground objects exhibit weak similarities to be grouped as a complete entity. Unlike TokenCut (Wang et al., 2023), which employs a graph cut based solely on attraction to isolate only the most distinctive parts of objects, such as the lampshade, we construct a joint weighted graph that incorporates both attraction and repulsion ($\boldsymbol{A}$&$\boldsymbol{R}$). This enables a more comprehensive segmentation to extract whole objects from the scene. Notably, it is the mutual repulsion ( highlighted by magenta dashed boxes) against the background that facilitates the segmentation of the *table lamp* and the *right vase* together.

descriptive parts of objects. In contrast, we adopt pairwise attraction and repulsion in a joint weighted graph to localize and segment whole objects.

**Unsupervised Video Segmentation.** Unsupervised video segmentation methods utilize abundant unlabeled videos as the source of free supervision (Ye et al., 2022; Wang et al., 2021; Yang et al., 2019a; Liu et al., 2021). The key ingredient is that motion across adjacent video frames provides useful constraints as training signals, such that elements tend to be perceived as a group if they move similarly. However, to guarantee the reliable motion information, these methods require externally supervised motion estimation networks (Teed & Deng, 2020; Sun et al., 2018), thus limiting their scalability. Although AMD (Liu et al., 2021) directly decomposes video sequences into regional segmentation and motion in an end-to-end manner, the characterization of regional motion often includes overly smoothed moving pieces and has difficulties in capturing fine details of object boundaries. However, our method requires neither optical flow as input or network training. Yet, our method demonstrate strong zero-shot segmentation on video data.

**Segmentation as Graph Cuts.** Normalized cut (Shi & Malik, 2000) presents image segmentation as a graph partitioning problem. It finds a grouping that maximizes the similarities within the partitions. (Ng et al., 2001) performs decomposition of the Laplacian matrix of a graph and then obtain the partitions by stacking the eigenvectors along the channel dimension. (Yu & Shi, 2001) studies perceptual popout using both feature similarity and local feature contrast. Objectness is popped out by measuring attraction and repulsion in a unified weighted graph. We harness the characteristics of joint measurement of attraction and repulsion for unsupervised whole object discovery.

## 3 UNSUPERVISED WHOLE OBJECTNESS BY CONTEXTUAL GROUPING

With the features from self-supervised visual transformers, we introduce the concept of attraction and repulsion from the feature similarity matrix and construct a joint weighted graph by both attraction and repulsion. We seek the graph cuts that maximize within-group attraction and figure-ground repulsion. We investigate the graph cuts not only within a single image, but also across related images in a co-segmentation setting. Contextual grouping with repulsion between similar images brings additional power for whole object discovery.

### 3.1 ATTRACTION AND REPULSION

Given an image $x$, we first extract its features $\boldsymbol{F}$ from self-supervised ViTs. A weighted graph is constructed where the nodes represent image square patches and the edges between nodes are defined as pairwise feature similarity. Let $\mathbb{V}$ denote the entire node set, and $\mathbb{V}_1, \mathbb{V}_2$ represent two disjoint subsets: $\mathbb{V}_1 \cup \mathbb{V}_2 = \mathbb{V}, \mathbb{V}_1 \cap \mathbb{V}_2 = \emptyset$. The cosine feature similarity $\boldsymbol{S}$ is: $\boldsymbol{S}(i,j) = \frac{<\boldsymbol{F}_i, \boldsymbol{F}_j>}{\|\boldsymbol{F}_i\|\|\boldsymbol{F}_j\|}$, where $\boldsymbol{F}_i$ and $\boldsymbol{F}_j$ denote the feature vectors of image patches $i$ and $j$ respectively. Intuitively, attraction between nodes characterizes how much two nodes attract each other to the same (unknown) group, assigning a larger weight for larger feature similarity. Similarly, repulsion measures how much

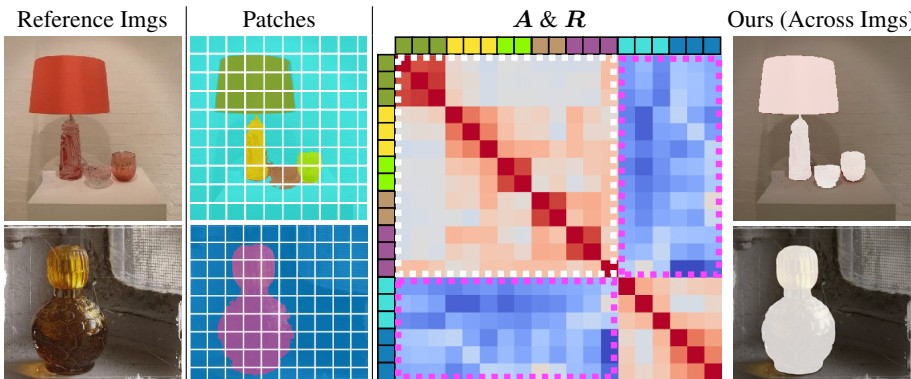

Figure 3: **We extend the idea of attraction and repulsion into a co-segmentation setting**, where the two semantically similar images are segmented jointly by the attraction and repulsion within and across themselves. Such a set of images can be obtained by k-nearest neighbors on feature space or finding frames sampled from the video of the same scene. The contextual grouping with the foreground-background repulsion (indicated by magenta dashed boxes) across these two images brings additional power for segmenting out the complete foreground objects, such as the *two vases* and the *table lamp* in the upper image, along with the *glass bottle* in the lower image.

nodes repel each other into (unknown) figure/ground segregation, a larger weight for larger feature dissimilarity. Given the similarity matrix $S$, attraction and repulsion matrices $A$ and $R$ are defined as Gaussian functions of $S$ (Fig.A1 in supplementary). TokenCut (Wang et al., 2023) utilizes attraction from self-supervised ViTs for graph cut which can only segment out characteristic local regions, not whole objects. An example of illustrating how TokenCut segment object parts is in Fig. 2.

Instead of using normalized cut by using only attraction, we investigate whether attraction and repulsion can jointly contribute to popping out whole objects. More technical details of attraction and repulsion are in the supplementary 1.1. Given attraction $A$ and repulsion $R$, we follow (Yu & Shi, 2001) and conduct a binary segmentation by using a unified grouping criterion

$$\max \xi_{AR} = \frac{\text{within-group } A}{\text{total degree of } A \& R} + \frac{\text{between-group } R}{\text{total degree of } A \& R}. \tag{1}$$

According to (Yu & Shi, 2001), the joint attraction and repulsion criterion is equivalent to

$$\max \xi_{AR}(\boldsymbol{p}) = \sum_{u=1}^{2} \frac{\boldsymbol{p}_u^T \boldsymbol{W} \boldsymbol{p}_u}{\boldsymbol{p}_u^T \boldsymbol{D} \boldsymbol{p}_u}, \tag{2}$$

$$\boldsymbol{W} = \boldsymbol{A} - \boldsymbol{R} + \boldsymbol{D}_R, \ \boldsymbol{D} = \boldsymbol{D}_A + \boldsymbol{D}_R,$$

where $\boldsymbol{p}_u$ is a binary membership vector for $\mathbb{V}_u$, $\boldsymbol{D}_A = \texttt{diag}(\texttt{sum}(\boldsymbol{A}, \texttt{dim} = 1)), \boldsymbol{D}_R = \texttt{diag}(\texttt{sum}(\boldsymbol{R}, \texttt{dim} = 1))$. The real valued solution to this partition problem is finding the second largest eigenvector $\mathbf{z}^*$ of the eigensystem

$$\boldsymbol{D}^{-1} \boldsymbol{W} \boldsymbol{z} = \lambda \boldsymbol{z}. \tag{3}$$

To illustrate our graph cut on joint weighted graph by attraction and repulsion, we show an example in Fig. 2. From the attraction and repulsion matrix in Fig. 2, the *lampshade*, *lamp base*, and *the vases* have weak similarities, thus using attraction is hard to bind them together as a whole object. It is the common repulsion of *the lamp* and *the right vase* against the background to bind them together.

### 3.2 CO-SEGMENTATION WITH ATTRACTION AND REPULSION

**1. Finding Reference Image Pairs.** So far we consider attraction and repulsion within a single image. It is straightforward to extend it to a co-segmentation setting, where two or more related images need to be jointly segmented. *These images should be semantically similar but visually distinct: If they are identical, no new information is gained for co-segmentation; if they are too dissimilar,*

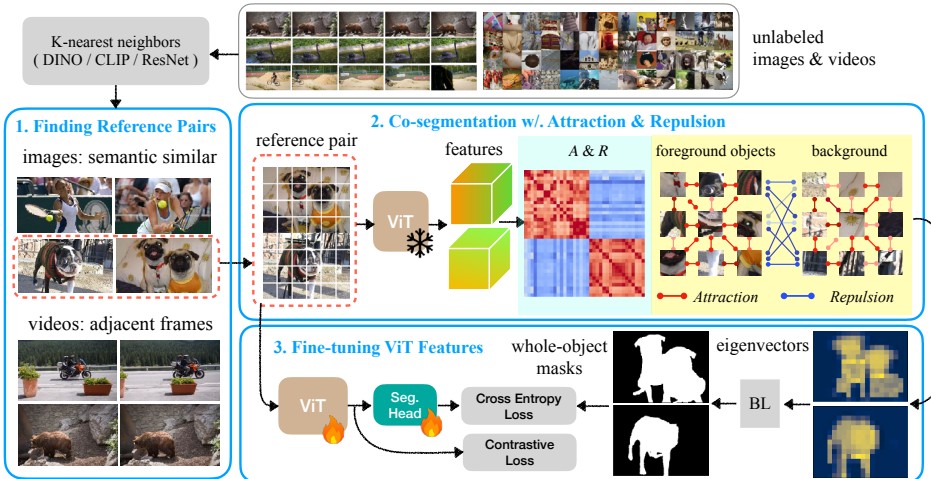

Figure 4: Our proposed framework for co-segmentation by attraction and repulsion. 1) We find reference pairs from the unlabeled images (that are semantically similar using K-nearest neighbors) or videos (using adjacent frames). 2) Given a reference pair, we discover the whole objects of them by the joint attraction and repulsion within and across themselves. The self-supervised ViT backbone is frozen at this stage. 3) With these whole object masks refined by the bilateral solver (BL) as supervision, we fine-tune the self-supervised ViT features along with the attached segmentation head.

*co-segmentation becomes ineffective.* Such a set of images could be obtained by k-nearest neighbors in some (e.g., DINO) feature space, computing visual embeddings from CLIP, or the video frames extracted from a video clip. Our examples of using CLIP to find reference image pairs from DUTS dataset (Wang et al., 2017) are shown in Fig. 5.

**2. Co-Segmentation with Attraction and Repulsion from Reference Images.** Given a pair of reference images $x_1$ and $x_2$, we first compute the self-supervised ViT features from the two images. Then, we concatenate these ViT features to build a joint graph containing the patches of both images as nodes, each patch node associated with its own feature vector. Our algorithm involves the following steps: 1) Compute the similarity matrix $S$; 2) Compute the attraction and repulsion matrix $A, R$ (according to the equations in Fig. A1); 3) Compute the matrices $D_A, D_R, W$, and $R$; 4) Find the 2nd largest eigenvector $z^*$ by solving the eigensystem at Equation 3. We separate $z^*$ into two vectors $z_1^*$ and $z_2^*$ to segment foreground objects in $x_1$ and $x_2$ simultaneously. An example of segmentation by attraction and repulsion across images is shown in Fig. 3; 5) To effectively segment objects based on $z^*$, we use the averaged value of $z^*$ for image partitioning and find out the part covering maximum absolute value from $z^*$ as the foreground objects.

**3. Fine-tuning Self-Supervised Features.** With the eigenvectors by attraction and repulsion to pop out whole object masks, we refine these masks with bilateral solver (Barron & Poole, 2016). Note that the bilateral solver is only applied during our fine-tuning stage. Furthermore, we utilize these whole object masks to fine-tune the self-supervised transformer features along with a lightweight segmentation head (1 `conv` $1 \times 1$ layer) with the cross-entropy and the contrastive loss. The cross-entropy loss is used to update the self-surprised features using whole object masks and the contrastive loss loss is to minimize foreground feature distances and maximize the foreground-background feature distances. The diagram for whole object discovery and self-supervised features fine-tuning is shown in Fig. 4.

## 4 EXPERIMENT

The evaluation of our methods for unsupervised whole object discovery is conducted on three tasks: unsupervised object discovery, unsupervised saliency detection, and unsupervised video object segmentation. The results of our method CGR-s are generated from *attraction & repulsion within a single image* while the results of our method CGR-c are produced by utilizing *attraction & repulsion*

Table 1: **CGR-s surpasses existing methods for unsupervised object discovery task**. In the setting of *w/o. learning* (no fine-tuning required), both CGR-s and CGR-c using attraction & repulsion outperform the *SoTA* method TokenCut (performance gap in blue) that use only attraction for object discovery in all three datasets. It shows that *attraction & repulsion can contribute together to localize whole objects in an unsupervised way*. With initial predictions by attraction and repulsion, both CGR-s and CGR-c involving self-supervised feature fine-tuning outperform the *SoTA* method HEAP (performance gap in green) in the setting of *w/. learning* (fine-tuning required).

Table 2: **CGR-s is a strong object segmenter for unsupervised video object segmentation task.** In *w/o. learning*, CGR-s considers attraction and repulsion within a single video frame outperforming TokenCut (performance gap in blue) which requires optical flow as input. Specifically, CGR-c considering attraction and repulsion across frames further improves the segmentation results on video sequences. In *w/. learning* setting, both CGR-s and CGR-c involving self-supervised feature fine-tuning outperform MG (the performance gaps are indicated in green). This shows that *CGR is a strong zero-shot object segmenter, utilizing attraction and repulsion to pop out whole objects, without requiring optical flow information as input.*

| Method | VOC07 | VOC12 | COCO20K |
|---|---|---|---|
| *w/o. Learning*, S/16-ViT | | | |
| DINO-seg (Caron et al., 2021) | 45.8 | 46.2 | 42.0 |
| LOST (Siméoni et al., 2021) | 61.9 | 64.0 | 50.7 |
| DSS (Melas-Kyriazi et al., 2022) | 62.7 | 66.4 | 52.2 |
| TokenCut (Wang et al., 2023) | 68.8 | 72.1 | 58.8 |
| CGR-s | 71.4 (+2.6) | **73.8** (+1.7) | 60.3 (+1.5) |
| CGR-c | **72.3** (+3.5) | 73.7 (+1.6) | **61.7** (+2.9) |
| *w/. Learning*, S/8-ViT | | | |
| SelfMask (Shin et al., 2022) | 72.3 | 75.3 | 62.7 |
| FOUND (Siméoni et al., 2023) | 72.5 | 76.1 | 62.9 |
| PEEKABOO (Zunair & Hamza, 2024) | 72.7 | 75.9 | 64.0 |
| HEAP (Zhang et al., 2024) | 73.2 | 77.1 | 63.4 |
| CGR-s | 76.4 (+3.2) | 79.8 (+2.7) | 65.6 (+2.2) |
| CGR-c | **77.7** (+4.5) | **80.8** (+3.7) | **66.2** (+2.8) |

| Method | Flow | Performance | | |
|---|---|---|---|---|
| | | DAVIS | FBMS | SegTV2 |
| *w/o. Learning* | | | | |
| TokenCut (Wang et al., 2023) | ✓ | 64.3 | 60.2 | 59.6 |
| CGR-s | ✗ | 66.4 (+2.1) | 62.5 (+2.3) | 61.2 (+1.6) |
| CGR-c | ✗ | **67.9** (+3.6) | **64.1** (+3.9) | **62.1** (+2.5) |
| *w/. Learning* | | | | |
| AMD (Liu et al., 2021) | ✗ | 45.7 | 28.7 | 42.9 |
| CUT (Keuper et al., 2015) | ✓ | 55.2 | 57.2 | 54.3 |
| FTS (Papazoglou & Ferrari, 2013) | ✓ | 55.8 | 47.7 | 47.8 |
| ARP (Koh & Kim, 2017) | ✓ | 76.2 | 59.8 | 57.2 |
| ELM (Lao & Sundaramoorthi, 2018) | ✓ | 61.8 | 61.6 | - |
| MG (Yang et al., 2021a) | ✓ | 68.3 | 53.1 | 58.2 |
| CGR-s | ✗ | 70.2 (+1.9) | 65.3 (+12.2) | 63.6 (+5.4) |
| CGR-c | ✗ | **71.4** (+3.1) | **65.8** (+12.7) | **64.5** (+6.3) |

*across reference images*. We consider two experimental settings: *w/o. learning* does not allow network training so we don't apply fine-tuning (or bilateral solver) for CGR-s and CGR-c ; *w/. learning* allows extra network training and we fine-tune for both CGR-s and CGR-c .

**Implementation Details.** We utilize the self-supervised features from DINO (Caron et al., 2021). We choose ViT-S/16 as the architecture for evaluation with the baselines in *w/o. learning* setting and ViT-S/8 to compare with the baselines in *w/. learing* setting. To find semantically similar but visually distinct images as reference images, we extract the CLS-token as the feature from DINO and run k-nearest neighbors. It takes less than 1 hour to run k-nearest neighbors as a preprocessing step. For videos, we use a frame interval of 10 to create reference image pairs for co-segmentation: [(00.jpg, 10.jpg), (01.jpg, 11.jpg), (02.jpg, 12.jpg), $\cdots$]. The repulsion weight $\omega$ is set to 0.2. The segmentation head contains a single conv $1 \times 1$ layer. During fine-tuning, we set the batch size to 4 and have 100 training epochs. The training is run on a single A40 NVIDIA GPU.

## 4.1 UNSUPERVISED OBJECT DISCOVERY

**Datasets & Eval Metric.** This task aims to identify entire objects within an image by specifying correct object bounding boxes. We use three widely recognized benchmarks: VOC07 (Everingham et al., 2010) containing 5011 images (train: 3507, val: 752, test: 752), VOC12 (Everingham & Winn, 2012) that includes 11540 images in total (train: 8078, val: 1731, test: 1731), and COCO20K (Vo et al., 2020) consisting of 19,817 images (train: 13873, val: 2972, test: 2972). Following the evaluation protocol (Wei et al., 2019; Cho et al., 2015), results are reported using the correct localization (*CorLoc*) metric, which measures the percentage of images where objects are correctly localized.

**Baselines.** In the setting of *w/o. learning*, we evaluate the results from CGR-s and CGR-c without self-supervised feature fine-tuning and directly compare with current methods that do not require network training, including DINO-seg (Caron et al., 2021), DSS (Melas-Kyriazi et al., 2022), LOST (Siméoni et al., 2021), and TokenCut (Wang et al., 2023). In the setting of *w/. learning*, we compare CGR-s and CGR-c involving fine-tuning with the methods SelfMask (Shin et al., 2022), FOUND (Siméoni et al., 2023), PEEKABOO (Zunair & Hamza, 2024), and HEAP (Zhang et al., 2024) that require network training or additional module learning.

Table 3: **CGR outperforms existing methods for unsupervised saliency detection task**. In the *w/o. learning* setting (no network training), CGR , utilizing both attraction and repulsion mechanisms, outperforms the *SoTA* method TokenCut which relies solely on attraction, across all three datasets (performance gap in blue). This indicates that the *combined use of attraction and repulsion promotes the segmentation of whole objects from unlabeled images*. Furthermore, in the *w/. learning* setting (training is required), with initial object masks by attraction and repulsion, CGR demonstrates superior performance by employing self-training with a lightweight module, surpassing the *SoTA* method HEAP (performance gap in green).

| Method | ViT | ECSSD | | | DUTS-TE | | | DUT-OMRON | | |
|---|---|---|---|---|---|---|---|---|---|---|
| | | maxF$_\beta$ | IoU | Acc. | maxF$_\beta$ | IoU | Acc. | maxF$_\beta$ | IoU | Acc. |
| *w/o. Learning* | | | | | | | | | | |
| BigBiGAN (Voynov et al., 2021) | – | 78.2 | 67.2 | 89.9 | 60.8 | 49.8 | 87.8 | 54.9 | 45.3 | 85.6 |
| FUIS (Melas-Kyriazi et al., 2021) | – | – | 71.3 | 91.5 | – | 52.8 | 89.3 | – | 50.9 | 88.3 |
| LOST (Siméoni et al., 2021) | S/16 | 75.8 | 65.4 | 89.5 | 61.1 | 51.8 | 87.1 | 47.3 | 41.0 | 79.7 |
| DSS (Melas-Kyriazi et al., 2022) | – | – | **73.3** | – | – | 51.4 | – | – | **56.7** | – |
| TokenCut (Wang et al., 2023) | S/16 | 80.3 | 71.2 | 91.8 | 67.2 | 57.6 | 90.3 | 60.0 | 53.3 | 88.0 |
| CGR-s | S/16 | 82.7(+2.4) | 72.8(+0.6) | 93.1(+1.3) | **69.5**(+2.3) | 60.2(+2.6) | 92.8(+2.5) | 62.6(+2.6) | 55.3(+2.0) | **90.7**(+2.7) |
| CGR-c | S/16 | **83.1**(+2.8) | 73.2(+2.0) | **94.7**(+2.9) | 69.3(+2.1) | **60.5**(+2.9) | **93.2**(+2.9) | **63.3**(+3.3) | 56.4(+3.1) | 90.6(+2.6) |
| *w/. Learning* | | | | | | | | | | |
| SelfMask (Shin et al., 2022) | S/8 | – | 78.1 | 94.4 | – | 62.6 | 92.3 | – | 58.2 | 90.1 |
| FOUND (Siméoni et al., 2023) | S/8 | **95.5** | 80.7 | 94.9 | 71.5 | 64.5 | 93.8 | 66.3 | 57.8 | 91.2 |
| PEEKABOO (Zunair & Hamza, 2024) | S/8 | 95.3 | 79.8 | 94.6 | **86.0** | 64.3 | 93.9 | **80.4** | 57.5 | 91.5 |
| HEAP (Zhang et al., 2024) | S/8 | 93.0 | 81.1 | 94.5 | 75.7 | 64.4 | 94.0 | 69.0 | 59.6 | 92.0 |
| CGR-s | S/8 | 94.1(+1.1) | 83.6(+2.5) | 95.2(+0.7) | 78.0(+2.3) | 65.9(+1.5) | **94.6**(+0.6) | 70.7(+1.7) | 60.8(+1.2) | 93.5(+1.5) |
| CGR-c | S/8 | 94.5(+1.5) | **83.9**(+2.8) | **95.8**(+1.3) | 78.2(+2.5) | **66.5**(+2.1) | 94.4(+0.4) | 71.2(+2.2) | **61.3**(+1.7) | **93.8**(+1.8) |

**Results.** The results are presented in Table 1. In *w/o. learning* setting, both CGR-s and CGR-c outperform TokenCut using the same `ViT-S/16` architecture. The performance gaps with TokenCut are noted in blue in Table 1. While TokenCut adopts graph cut by using only attraction, TokenCut could only localize discriminative parts of objects in unlabeled images. In contrast, both CGR-s and CGR-c localize objects from unlabeled images by a weighted graph of combining attraction and repulsion. It demonstrates that *attraction and repulsion can contribute together in a joint weighted graph to pop out whole objects for unsupervised object discovery*. In *w/. learning* setting, by fine-tuning the self-sueprvised features, CGR-s and CGR-c present higher scores than the current *SoTA* model HEAP using the same `ViT-S/8` architecture. The performance gaps with HEAP are presented in green in Table 1. This is to show that *with the initial whole object predictions by attraction and repulsion, fine-tuning the self-supervised features enhances whole object localization and helps achieve new state-of-the-art performance* in unsupervised object discovery task.

## 4.2 Unsupervised Saliency Detection

**Datasets** & **Eval Metrics.** We consider three datasets: ECSSD (Shi et al., 2015) containing 1000 images (train: 700, val: 150, test: 150), DUT-OMRON (Yang et al., 2013) including 5186 images (train: 3630, val: 778, test: 778), and DUTS (Wang et al., 2017) with 1580 images (train: 7373, val: 1580, test: 1580). We adopt three standard metrics: mean intersection-over-union (IoU) with a threshold set at 0.5, pixel accuracy (Acc), and the maximal $F_\beta$ score (max $F_\beta$), where $\beta^2$ is set to 0.3, in accordance with Wang et al. (2023), Siméoni et al. (2023), and Zhang et al. (2024).

**Baselines.** In the setting of *w/o. learning*, we evaluate CGR-s and CGR-c without self-supervised feature fine-tuning and directly compare it with current methods that do not require network training. These methods include BigBiGAN (Voynov et al., 2021), FUIS (Melas-Kyriazi et al., 2021), LOST (Siméoni et al., 2021), DSS (Melas-Kyriazi et al., 2022), and TokenCut (Wang et al., 2023). In the setting of *w/. learning*, we conduct self-supervised feature fine-tuning on CGR-s and CGR-c and compare them with SelfMask (Shin et al., 2022), FOUND (Siméoni et al., 2023), PEEKABOO (Zunair & Hamza, 2024), and HEAP (Zhang et al., 2024) that necessitate network training or additional module learning.

**Results.** Our results on unsupervised saliency detection are shown in Table 3. In the *w/o. learning* setting, both CGR-s and CGR-c surpass TokenCut using the same `ViT-S/16` architecture. The performance differences with TokenCut are highlighted in blue in Table 3. TokenCut, which employs graph cut using only attraction, can only segment discriminative parts of objects in unlabeled images. In contrast, CGR identifies entire objects in unlabeled images by utilizing a weighted graph that combines attraction and repulsion. This demonstrates that *the combined use of attraction and*

| Ref Imgs | TokenCut | FOUND | Ours | Ref Imgs | TokenCut | FOUND | Ours |
| --- | --- | --- | --- | --- | --- | --- | --- |

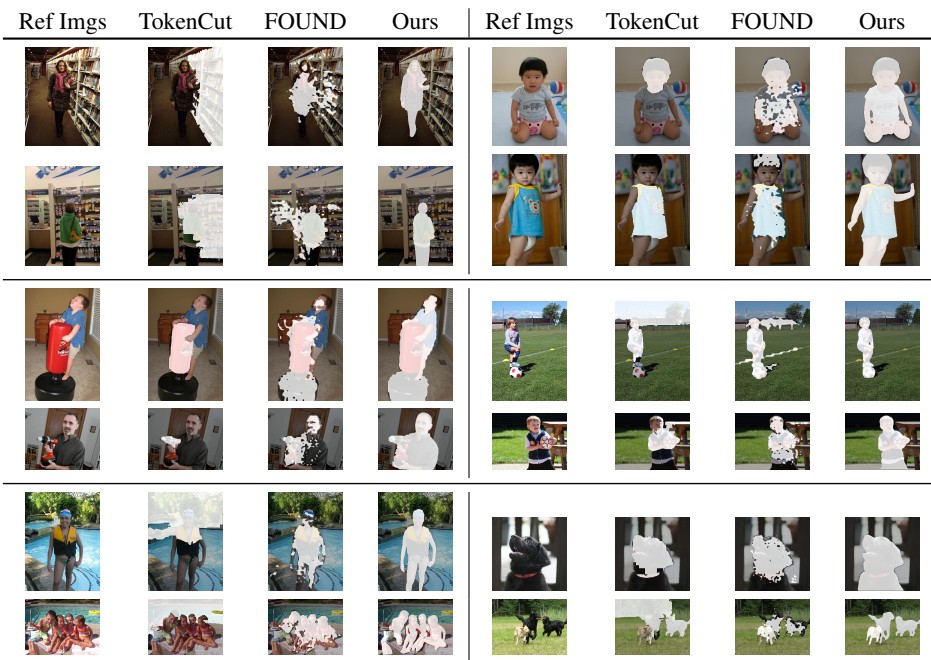

Figure 5: Our CGR-c is compared with TokenCut (*SoTA* in the *w/o. training* setting) and FOUND (*SoTA* in the *w/. training* setting) for unsupervised saliency detection. Both TokenCut and FOUND segment out only discriminative parts of the object using attraction. In contrast, our method applying attraction and repulsion within images and across reference images pops out the whole objects.

*repulsion promotes not only localization but also segmentation of whole objects*. In the *w/. learning* setting, CGR-s that leverages self-supervised feature fine-tuning achieves higher scores than the current state-of-the-art model HEAP using the same `ViT-S/8` architecture. The performance differences with HEAP are shown in green in Table 3. This indicates that *fine-tuning self-supervised features with initial object masks by attraction and repulsion greatly refines whole object segmentation and builds new SoTA performance on saliency detection*. The visualization results are shown in Fig. 5.

### 4.3 Unsupervised Video Object Segmentation

**Datasets & Eval Metric.** We conduct experiments on three widely recognized benchmarks for video object segmentation. These benchmarks include DAVIS (Perazzi et al., 2016) including 50 videos in total (train: 30, val: 10, test: 10), FBMS (Ochs et al., 2013) that has 59 videos (train: 25, test: 30), and SegTV2 (Li et al., 2013) containing 14 videos (train: 6, test: 7). We merge the annotations of all moving objects into a single mask for both the FBMS and SegTV2 datasets following (Wang et al., 2023; Yang et al., 2021c). We also test our method on CO3D dataset (Reizenstein et al., 2021). The performance is assessed using the Jaccard index ($\mathcal{J}$), which quantifies the intersection over union (IoU) between the predicted segmentation masks and the ground-truth annotations.

**Baselines.** CGR is evaluated against several unsupervised video object segmentation methods, many of which rely on optical flow information during the training phase. These methods include AMD (Liu et al., 2021), CUT (Keuper et al., 2015), FTS (Papazoglou & Ferrari, 2013), APR (Koh & Kim, 2017), ELM (Lao & Sundaramoorthi, 2018), MG (Yang et al., 2021a), and SAM2 (Ravi et al., 2024). Notably, AMD circumvents the need for optical flow by utilizing motion networks that predict motion information directly from unlabeled video frames. TokenCut which requires no network training still depends on optical flow as input data. SAM2 is supervisedly trained on large amounts of human annotations. In contrast, CGR-c operates without requiring optical flow or any human annotations.

**Results.** The results on unsupervised video object segmentation is in Table 2. In the *w/o learning* setting, CGR considers attraction and repulsion within a single video frame outperforming TokenCut

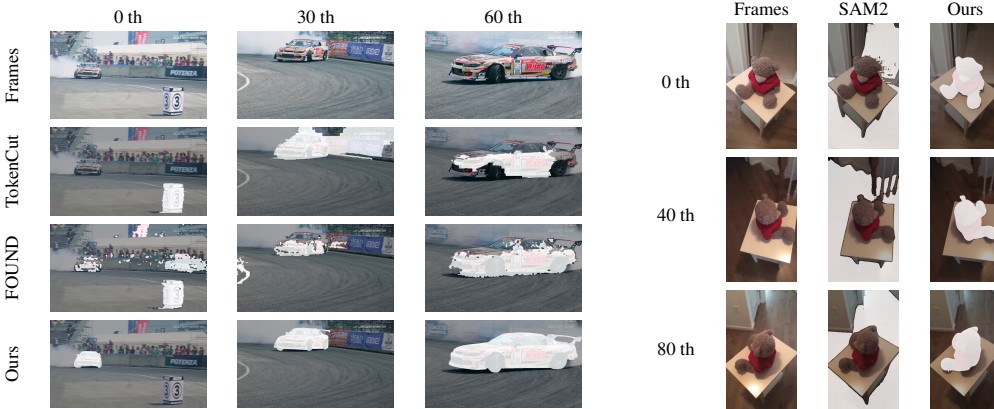

Figure 6: CGR-c is compared with TokenCut and FOUND for unsupervised video segmentation on DAVIS dataset. Both TokenCut and FOUND only using attractions in a single image fail to pop out the vehicle from the background, while CGR-c using related video frames (0th, 30th, 60th) as reference pairs is capable of segmenting the whole car body from the background.

Figure 7: CGR-c is compared with SAM2 for unsupervised video segmentation on CO3D dataset. SAM2 supervisedly pre-trained wrongly segments the floor as the foreground while CGR-c with reference frames pops out the teddybear off the background.

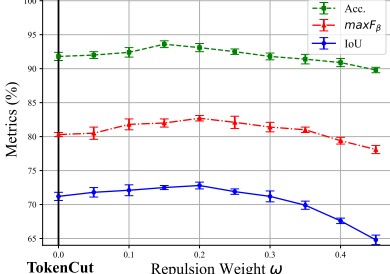

Figure 8: The unsupervised saliency detection performance of CGR-s on ECSSD dataset with different values of repulsion weight $\omega$. CGR is the same as TokenCut when $\omega = 0$ because the impact of repulsion is set to zero in grouping.

Figure 9: The performance of CGR-c with video frames at different video frame intervals for unsupervised video object segmentation. CGR-c is the equivalent to CGR-s when the video frame interval is 0.

(performance gaps are noted in blue). Moreover, CGR-c takes attraction and repulsion both within and across adjacent frames further boosting the video segmentation results. It shows that *CGR utilizing attraction and repulsion to pop out whole objects, is a strong zero-shot object segmenter from unlabeled video data, without requiring optical flow information as input.* The visualization of the results is shown in Fig. 6. We further compare CGR-c with SAM2 on the CO3D dataset in Fig. 7. Without any prompts, SAM2 fails to segment out the teddybear as the foreground object. *It highlights the effectiveness of CGR-c to pop out the whole objects using attraction and repulsion across adjacent video frames.*

### 4.4 ABLATION

**Repulsion Weight.** To adjust the relative importance between attraction and repulsion, we introduce a repulsion weight factor $\omega$, where $\omega \in [0, 1]$. We study the impact of different adjustments of repulsion weight $\omega$ on CGR-s for ECSSD unsupervised saliency detection in Fig. 8. The performance of TokenCut (Wang et al., 2023) is shown at the black vertical line ($\omega = 0$). When $\omega$ is set to 0, CGR is equal to TokenCut. The best performance for $\max F_{\beta}$, Acc. and IoU in ECSSD dataset (Shi et al., 2015) is achieved when $\omega$ is set to 0.2. We use this value of $\omega$ for all the other datasets during implementation so $\omega$ *does not need to be tuned at each experiment.*

| Ours | EigVec (Ours) | EigVec (TokenCut) | Ours | EigVec (Ours) | EigVec (TokenCut) |
|---|---|---|---|---|---|
| 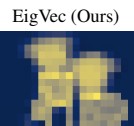 | 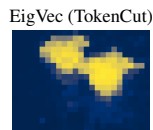 | 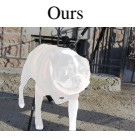 | 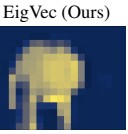 | 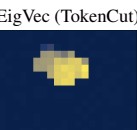 | |

Figure 10: The eigenvectors of CGR-c using attraction and repulsion across reference images pop out the whole body of the dogs while the eigenvectors of TokenCut utilizing attraction pop out only the head part of the dogs.

Table 4: The performance of CGR-c using reference images paired with different techniques. The results are evaluated on ECSSD dataset in the task of unsupervised saliency detection in the *w/o. training* setting.

| Technique | ECSSD | | |
|---|---|---|---|
| | $\text{maxF}_\beta$ | IoU | Acc. |
| DINO (Caron et al., 2021) | 83.1 | 73.2 | 94.7 |
| ResNet-50 (He et al., 2016) | 83.4 | **74.2** | 95.6 |
| CLIP (Radford et al., 2021) | **83.8** | 73.8 | **95.8** |

Table 5: The unsupervised saliency detection results on ECSSD dataset with the attached segmentation head using different architectures. These results are from CGR-c by fine-tuning the self-supervised features (*w/. training* setting).

| Arch | ECSSD | | |
|---|---|---|---|
| | $\text{maxF}_\beta$ | IoU | Acc. |
| $1 \times \text{Conv}(1,1)$ | 94.5 | 83.9 | 95.8 |
| $2 \times \text{Conv}(1,1)$ | **95.2** | **84.4** | **96.3** |
| $3 \times \text{Conv}(1,1)$ | 92.3 | 81.5 | 92.7 |

**Reference Image Discovery.** Our CGR-c utilizes attraction and repulsion across two similar images. Current methods include using K-Nearest Neighbors on the DINO's features, on the features extracted from ImageNet pre-trained models, or on the visual embeddings from CLIP model (Radford et al., 2021). We adopt DINO features in all the experiments as we want to reduce dependence on additional models. We conduct ablation study on searching similar image pairs as reference images using DINO features, ResNet-50 (pretrained on ImageNet) features, and CLIP model in Table 4.

**Reference Image Discovery for Videos.** CGR-c takes two frames from a video sequence as a pair of reference images. These two frames are possibly located at different timestamp in a video clip. We study the impact of different video frame intervals on unsupervised video object segmentation in Fig. 9. The video frame intervals between 8 to 18 yield better results according to our ablation. For all our experiments on unsupervised video object segmentation, we set up the frame interval with 10. When the video frame interval is set to 0, CGR-c is equivalent to CGR-s as the two reference images are the same – no new information brought up.

**Segmentation Head.** We further study fine-tuning along with the attached segmentation head under different architectures. Note that FOUDN utilize $\text{Conv}(1,1)$ as the segmentation head. Experimental results of different architectures for the segmentation head are shown in Table 5. The performance increases when applying $2 \times \text{Conv}(1,1)$ but drops by using $3 \times \text{Conv}(1,1)$.

**Eigenvectors.** As shown in Fig. 10, the eigenvectors of CGR-c using attraction and repulsion across reference images pop out the whole body of the dogs while the eigenvectors of TokenCut utilizing attraction pop out only the head of the dogs.

## 5 CONCLUSION

In this paper, we introduced a novel approach to unsupervised object discovery and segmentation using a spectral graph partitioning framework that harnesses both attraction and repulsion cues. Our method effectively segments whole objects by optimizing within-group attractions and minimizing distractions from the background. It significantly outperforms existing methods across benchmarks for unsupervised object discovery, figure/ground saliency detection, and video object segmentation. The simplicity and robustness of our approach make it a promising tool for advancing autonomous and robust visual perception in various applications, suggesting a significant step forward in the field of computer vision.

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
