# OpenReview forum: "Unsupervised Whole Object Discovery by Contextual Grouping with Repulsion"
_ICLR.cc/2025/Conference — ICLR 2025 Conference Withdrawn Submission_

### Official Review · Reviewer_aVoJ · 2024-11-02

**Soundness:** 2
**Presentation:** 3
**Contribution:** 2
**Rating:** 5
**Confidence:** 4

**Summary:**

The paper proposes a method to perform unsupervised object discovery and segmentation in videos. It utilizes spectral graph partitioning with both feature similarity and dissimilarity cues to capture whole objects from unlabeled images. A graph segmentation model is trained using cross-entropy loss and contrastive loss. The quality of segmentation on image and video datasets appears to improve compared to previous approaches.

**Strengths:**

The paper is well-written and easy to follow.
The proposed approach is an extension of the prior TokenCut approach, which utilized spectral graph partitioning with an attraction cue. Here, the method is extended by incorporating both attraction and repulsion cues in the graph structure, as proposed in Yu and Shi, 2001.
Moreover, the paper adapts the framework to video data by introducing multi-frame objectives.

**Weaknesses:**

However, given the main contributions, the paper appears to be incremental, with limited innovation beyond extending existing methods.
The method focuses on extracting a single dominant object in the scene, which wouldn't apply to complex scenes with many objects.
The results are primarily demonstrated on older datasets like VOC, COCO and DAVIS, and the comparisons focus largely on previous approaches, lacking evaluation against more recent research in the field, such as VideoCutLER: Surprisingly Simple Unsupervised Video Instance Segmentation (CVPR, 2004).

**Questions:**

1. The method relies on the initial segmentation provided by the graph cut method. How does it recover from large-scale errors in this prior segmentation.
2. How would this method be extended to multi-object segmentation.
3. How does it work with self-similar objects. E.g., multiple instances of the same object in the image.
4. How well does this method work on more complex datasets like YoutubeVIS etc.

---

### Official Review · Reviewer_mTRy · 2024-11-03

**Soundness:** 3
**Presentation:** 3
**Contribution:** 3
**Rating:** 6
**Confidence:** 2

**Summary:**

It is challenging to discover and segment whole objects from unlabeled images, as features unsupervisedly learned on images tend to focus on distinctive appearances (e.g., the face rather than the torso), and grouping by feature similarity could reveal only these representative parts, not the whole objects (e.g., the entire human body). The key insight of this paper is that, an object of distinctive parts pops out as a whole, due not only to how similar they are to each other, but also to how different they are from their contexts within an image or across related images. The latter could be crucial for binding different parts into a coherent whole without preconception of
objects. This paper formulate this idea for unsupervised object segmentation in a spectral graph partitioning framework, where nodes are patches and edges are grouping cues between patches, measured by feature similarity for attraction, and by feature dissimilarity for repulsion. This paper seek the graph cuts that maximize within-group attraction and figure-ground repulsion while minimizing figure/ground attraction and within-group repulsion. The  simple method consistently outperforms the state-of-the-art on unsupervised object discovery, figure/ground saliency detection, and unsupervised video object segmentation benchmarks. In particular, it excels at discovering whole objects instead of salient parts.

**Strengths:**

The strengths are as follows:
1. This formulate the idea "an object of distinctive parts pops out as a whole, due not only to how similar they are to each other" for unsupervised object segmentation in a spectral graph partitioning framework, where nodes are patches and edges are grouping cues between patches, measured by feature similarity for attraction, and by feature dissimilarity for repulsion.
2. This paper seek the graph cuts that maximize within-group attraction and figure-ground repulsion while minimizing figure/ground attraction and within-group repulsion.
3. This paper investigate this idea not only within a single image, but also across related images in a co-segmentation setting, where contextual grouping with repulsion between images brings additional power for discovering whole objects together

**Weaknesses:**

This paper present a method for unsupervised segmentation/saliency detection/co-segmentation. The weakness are as follows:
1. The time cost and memory consumption for the proposed method is not presented. This is quite necessary as the method use a large model like ViT.
2. What does the Self-Supervised Transformer indicate in Figure5? How about the segmentation head? does it use a pretrained sementation model? Looks like it use a mask from a segmentation model as gt to compute loss, right?
3. In figure 8, the paper try to compare the result with SAM2, but only a few visual results are provided, it there more systematic comparison results?

**Questions:**

No

---

> ### Author Response · Authors · 2024-12-02
>
> Thank you for your thorough review and valuable feedback. We have revised the submission, with updated sections highlighted in purple. We address your concerns in detail below.
>
> **1. Time and memory consumption of the proposed method**
>
> In our experiments, fine-tuning CGR with S/8 ViT architecture is conducted on 4 A40 GPUs for 3 days of training. CGR does not require or depend on extensive computational devices and large amounts of datasets.  Our CGR is computationally efficient compared with DINO, which requires more than 16 GPUs over 3 days of training on ImageNet.
>
> **2. Clarification on Figure 5**
>
> - **Self-supervised transformer**: This refers to the ViT backbone pre-trained using self-supervised learning. In our experiments, we utilize DINO pre-trained with self-distillation as the ViT backbone.
> - **Segmentation head**: The segmentation head contains a single 1x1 convolution layer.
> - **Ground-truth masks**: Yes, the masks from co-segmentation serve as ground truth to compute the loss.
>
> **3. Compared with SAM2 for video object segmentation on the DAVIS dataset**
> We compare the performance of segmenting foreground video objects on DAVIS datasets using SAM2 and our proposed CGR (CGR-c, co-segmentation setting). We use the mean intersection over the union between the predicted foreground masks and ground truth as the metric. The results are shown here:
> | Method | mIoU |
> |--------|------|
> | SAM2   | 69.8 |
> | CGR    | 71.4 |
> As a supervised pre-trained method, SAM2 is less effective than our CGR as SAM2 is confused between the foreground objects and background, while CGR using co-segmentation on reference frames pops out of the foreground off the background.

---

### Official Review · Reviewer_3kb7 · 2024-11-04

**Soundness:** 3
**Presentation:** 3
**Contribution:** 2
**Rating:** 6
**Confidence:** 3

**Summary:**

This paper addresses the unsupervised object segmentation task. The authors proposed the Contextual Grouping with Repulsion method (CGR), which considers both the internal similarities (attraction) among different parts of an object and their common dissimilarities (repulsion) to the background. The authors formulate their pipeline using a weighted graph where nodes represent image patches and edges encode grouping cues, measured by both feature similarity (attraction) and dissimilarity (repulsion). The proposed approach extends TokenCut, which solely relies on internal similarities between different object parts for segmentation. The proposed method demonstrates superior performance across multiple unsupervised segmentation benchmarks, including unsupervised object discovery, saliency detection, and video object segmentation.

**Strengths:**

1.	The proposed CGR is simple and easy to understand.
2.	The paper is well-written and organized, making the author's ideas easy to understand.
3.	The authors validated CGR's performance on different segmentation benchmarks.

**Weaknesses:**

1.	This paper lacks sufficient details about the training and evaluation process. Specifically, it does not explain how the train/validation/test sets were divided and which data subset was used in the training, hyperparameter selection, and final model evaluation.
2.	Regarding the repulsion weight, Figure 9 shows that when $\omega$ fluctuates in the range of 0~0.25, the performance difference is not significant, which raises doubts about the effectiveness of the proposed method. Additionally, the author only conducted an ablation study of $\omega$ on the ECSSD dataset for unsupervised saliency detection and then applied this parameter to all tasks and datasets. I suppose this pattern is not convincing enough. I'm not suggesting that the authors should conduct ablation studies for all tasks to determine the repulsion weight. Rather, I think it's tricky to set this parameter as a fixed value and apply it to different tasks and datasets. The authors should discuss whether this parameter could adapt automatically when facing different tasks and datasets.
3.	Still for the repulsion weight, comparing the experimental results, it appears that the authors used the same data subset for both hyperparameter selection (Figure 9) and results reporting (Table 3). In other words, the authors did not strictly distinguish between the validation set and test set in the experiments, which suggests that their proposed method might be overfitting to the target dataset.

**Questions:**

My main concerns are about the training/evaluation process and parameter selection. Please refer to the weakness section.

---

### Official Review · Reviewer_BDiv · 2024-11-04

**Soundness:** 2
**Presentation:** 3
**Contribution:** 2
**Rating:** 3
**Confidence:** 3

**Summary:**

The paper proposes a solution of discovering and segmenting objects in unsupervised setting. Inspired by object feature similarity as well as feature disimilarity, the paper proposes to utilize graph cuts that maximize similarity between object features while also maximizing dissimilarity between object and background features. Moreover, the paper shows performance gain for unsupervised object
discovery, saliency detection, and unsupervised object detection

**Strengths:**

The main motivation of the proposed method is to focus on distinctive parts of an object by increasing similarity between them and simultaneously focus on how dissimilar they are from their context in the image. The paper first identifies this problem in existing method and show that upon taking into account both similarity and dissimilarity, there is a possibility of performance improvement. Empirically for three different unsupervised tasks, the proposed method show improvements over existing methods in both single image setting and reference image setting.

**Weaknesses:**

The proposed idea of utilizing attraction and repulsion doesn’t seem to be novel, As authors say in L194 “Given attraction A and repulsion R, we follow (Yu & Shi, 2001 ) and conduct. . . ”. The referenced paper proposes the same idea of utilizing attraction and repulsion for both to measure degree of attraction and segregation of features. The difference seems to be application of this on features obtained from self-supervised transformers instead of image features. Moreover, the segmentation method remains the same as before. The rest of the method is clearly followed from (Wang et al, 2023).

There are also concerns regarding reported quantitative results in table 3. As mentioned in L257, the authors use bilateral solver (BL) to refine the masks. However, when comparing with TokenCut (Wang et al, 2023), the results are taken without bilateral solver, TokenCut+BL shows better performance by a significant margin when compare with CGR (proposed method). Similarly there is inconsistency in table 2. TokenCut+BL is not reported, which clearly outperforms CGR.

Another minor concern in the paper is repetitive writing. There are multiple instances in abstract and introduction where sentences are
repeated again and again. For e.g. L19 and L50. Also, few argumentative sentences in the paper are too long and complex which hinders the information being conveyed. This should be improved for clear understanding of the paper.

**Questions:**

1. Could you please clearly emphasize the novelty part and distinguish this work from the existing literature?
2. Could you please clarify how the comparisons drawn in the results are fair?

---

### Note · Authors · 2025-03-07

I have read and agree with the venue's withdrawal policy on behalf of myself and my co-authors.

---

### Meta-Review · Area_Chair_yHip · 2024-12-11

**Metareview:**

This paper aims to discover whole object by contextual grouping with repulsion.  Specifically, the key insight of this paper is that an object of distinctive parts pops out as a whole, due not only to how similar they are to each other, but also to it how different they are from their contexts within an image or across related images. The latter could be crucial for binding different parts into a coherent whole without preconception of objects. Therefore, the authors formulate their idea for unsupervised object segmentation in a spectral graph partitioning framework, where nodes are patches and edges are grouping cues between patches, measured by feature similarity for attraction, and by feature dissimilarity for repulsion. Experimental results reveal that the proposed method consistently outperforms the state-of-the-art algorithms on unsupervised object discovery, figure/ground saliency detection, and unsupervised video object segmentation benchmarks.

The paper is generally written clearly, and the key insight of this paper is also reasonable. However, the review committee considers that the main idea of this paper is not sufficiently novel for ICLR.

**Additional Comments On Reviewer Discussion:**

One of the reviewer raises his/her score after rebuttal. The reviewer considers that the idea of "whole object" is fuzzy. Self-similar objects should be recognized as individual instances, and not as a collective blob. Also, the spectral graph partitioning has also been investigated before, and thus not compelling enough to be an ICLR submission.

Due to the above reasons, as well as that the score of this paper is not high in my AC pool, I feel sorry that I cannot recommend an acceptance due to the limited acceptance rate.

---

### Decision · Program_Chairs · 2025-01-22

Reject